# Efficient Conformer for Agglutinative Language ASR Model Using Low-Rank Approximation and Balanced Softmax

**Ting Guo, Nurmemet Yolwas * and Wushour Slamu**

Xinjiang Multilingual Information Technology Laboratory, Xinjiang Multilingual Information Technology Research Center, College of Information Science and Engineering, Xinjiang University, Urumqi 830017, China
* Correspondence: nurmemet@xju.edu.cn

**Abstract:** Recently, the performance of end-to-end speech recognition has been further improved based on the proposed Conformer framework, which has also been widely used in the field of speech recognition. However, the Conformer model is mostly applied to very widespread languages, such as Chinese and English, and rarely applied to speech recognition of Central and West Asian agglutinative languages. There are more network parameters in the Conformer end-to-end speech recognition model, so the structure of the model is complex, and it consumes more resources. At the same time, we found that there is a long-tail problem in Kazakh, i.e., the distribution of high-frequency words and low-frequency words is not uniform, which makes the recognition accuracy of the model low. For these reasons, we made the following improvements to the Conformer baseline model. First, we constructed a low-rank multi-head self-attention encoder and decoder using low-rank approximation decomposition to reduce the number of parameters of the multi-head self-attention module and model's storage space. Second, to alleviate the long-tail problem in Kazakh, the original softmax function was replaced by a balanced softmax function in the Conformer model; Third, we use connectionist temporal classification (CTC) as an auxiliary task to speed up the model training and build a multi-task lightweight but efficient Conformer speech recognition model with hybrid CTC/Attention. To evaluate the effectiveness of the proposed model, we conduct experiments on the open-source Kazakh language dataset, during which no external language model is used, and the number of parameters is relatively compressed by 7.4% and the storage space is relatively reduced by 13.5 MB, while the training speed and word error rate remain basically unchanged.

**Keywords:** efficient Conformer; Kazakh; low-rank decomposition; multi-head self-attention; balanced softmax

## 1. Introduction

Automatic speech recognition (ASR) is capable of automatically converting speech signals into corresponding text sequences. With the continuous iterative development of deep learning techniques, end-to-end speech recognition models are gradually replacing the traditional Hidden Markov Model (HMM)-based speech recognition models. End-to-end speech recognition models are favored by more and more researchers because they simplify the overall model structure and training steps, and reduce the reliance on domain knowledge compared to traditional HMM-based speech recognition models. The most typical representative of end-to-end speech recognition models is Transformer [1], which uses encoders and decoders with a multi-head attention mechanism to simplify the training and recognition process, accelerate parallel computation, and improve the accuracy of model recognition. Recently, Gulati et al. [2] proposed the Conformer model, which uses a convolutional module to capture local content dependencies. Its shows better results on English datasets [3] compared to other models such as Transformer and Transformer-XL [4].

Although the Conformer end-to-end models have shown better results in the field of speech recognition and are widely used, they have also caused their own number

of parameters to become larger and larger compared to Transformer. Some Conformer-based end-to-end speech recognition models have reached tens of millions of parameters, which also brings about problems such as higher computational complexity of the model, more resource consumption and large storage space. Therefore, it is important to build a lightweight end-to-end speech recognition model.

Many works have been carried out to study this. Winata et al. [5] constructed the low-rank Transformer (LRT) using low-rank multi-head attention and low-rank feed-forward. Wang et al. [6] used sparse processing for the computation of self-attention in the Conformer model to reduce the computational complexity on attention. In addition, MHA was used to carry out down sampling to improve the performance of the Conformer model [7]. Li et al. constructed a low-rank feed-forward network module and a multi-head linear attention module to reduce the model parameters for the Conformer model [8]. Ref. [9] uses two approaches based on singular value decomposition (SVD) to solve the adaptation and personalization problems of DNN. Low-rank compression can compress the parameters of the model, but at the same time, it has an impact on the recognition accuracy of the model; moreover, to our best knowledge, few research works have investigated compression for Conformer's multi-headed attention module.

There is a long-tail problem in most data sets. The long-tail problem has been extensively studied in computer vision [10,11]; for example, Tang et al. [12] used causality to construct a balanced classifier to solve the long-tail problem in vision. Additionally, we found that there is a long-tail problem in the open-source Kazakh language dataset [13], that is, high-frequency words occupy a small part of the dataset, while low-frequency words occupy the majority of the dataset. The uneven distribution of high-frequency words and low-frequency words makes the final recognition accuracy higher for high-frequency words that are adequately trained and lower for low-frequency words that are not adequately trained. At the same time, some low-frequency words may be incorrectly classified and recognized because of speech similarity, resulting in the overall low recognition accuracy of the final model [14,15]. To the best of our knowledge, few studies have explored the long tail in agglutinative languages.

To address the above issues, we focus on constructing a lightweight but efficient Conformer speech recognition model, using low-rank approximation decomposition to reduce the number of redundant parameters of the multi-head self-attention module in the Conformer model and the memory size of the model, while keeping the model recognition accuracy and training time largely unchanged. The main innovations in this article are as follows. Firstly, we construct an encoder and decoder of low-rank multi-head self-attention modules, called LMHSA modules, in the Conformer model, and do not need to retrain the overall model during the compression process. Compression of the multi-head attention module using low-rank decomposition reduces the number of redundant parameters and the memory size of the model. Second, to effectively alleviate the long-tail problem in Kazakh, we used a softmax function with penalty factor to replace the original softmax function, called balanced softmax, to improve the accuracy of the model without increasing the training difficulty. Meanwhile, we investigate the best value of penalty factor in the Conformer model. Third, we use CTC as an auxiliary function in the Conformer model to build a hybrid CTC/Attention multi-task-learning training approach to help the model converge quickly. Fourth, we build a lightweight but efficient Conformer model, reducing the number of parameters and the storage space of the model while keeping the training speed and recognition accuracy largely unchanged. Experimental results on the KSC dataset show that the use of an LMHSA module can effectively reduce the parameters of Conformer, and the number of parameters of the model is reduced by 7.4%. The use of the balanced softmax function can effectively alleviate the long-tail problem in the Kazakh language, and the speech recognition word error rate (WER) is reduced by 0.15%. The multi-task efficient Conformer model using hybrid CTC/Attention compresses the final number of parameters of the model by 7.4% and the storage space of the model by 13.5 MB, while the overall training speed and word error rate remain largely unchanged.

The rest of this paper is as follows. In Section 2, we briefly introduce the hybrid CTC/Attention end-to-end speech recognition model and its application to Kazakh, while we introduce the decomposition and the speech long-tail problem. The proposed method is described in detail in Section 3. Section 4 presents the environment configuration of the experiments and the introduction of the dataset. Section 5 presents the experiments conducted with our proposed low-rank multi-head self-attention encoder and decoder, balanced softmax, and the final model structure scheme compared to the baseline model Conformer. In Section 6, we summarize the work we have completed and present the outlook.

## 2. Related Work

End-to-end automatic speech recognition models are increasingly being investigated by researchers due to their simplified training process and time, and their ability to effectively improve the recognition accuracy of models. Currently, the mainstream end-to-end speech recognition models mainly include connectionist temporal classification (CTC) [16,17], the recurrent neural network transducer (RNN-T) model [18,19], and the attention mechanism-based encoder and decoder model [20,21]. Among the attention-based mechanism models, the most representative one is Transformer [1]. Transformer uses a multi-head attention mechanism, increases parallel computation, and achieves good results in sequence modeling. Transformer, although more effective in extracting long sequence dependencies, performs poorly in extracting local features. For this reason, Gulati et al. proposed a new model, Conformer, which is a convolutional module to enhance Transformer's learning in local features [2]. The proposed end-to-end speech recognition framework based on Conformer has further improved the recognition accuracy of end-to-end speech recognition models and has also been widely used in the field of speech recognition [22,23]. The end-to-end model based on the attention mechanism is difficult to train due to its overly flexible alignment. Therefore, adding CTC to the decoder layer of the attention-based mechanism to assist in training not only helps the model converge quickly, but also improves the recognition rate of the model [24,25].

With the continuous development of deep learning techniques, end-to-end speech recognition is becoming more and more popular among researchers, but it also increases the number of parameters and storage space for models, which leads to higher resources and a longer training time for end-to-end speech recognition models. There are many research works investigating the reduction in the number of redundant parameters in end-to-end speech recognition models. Winata et al. proposed a method to compress large pre-trained models using non-negative matrix decomposition LSTM post-training compression [26]. Kriman et al. proposed a QuartzNet model with one-dimensional temporal channel separable convolution to decompose the convolution [27]. Mehrotra et al. proposed to compress the end-to-end speech recognition model using a low-rank decomposition method, with no reduction in word error rate [28]. Winata et al. [5] constructed a lightweight but effective end-to-end speech recognition model using low-rank decomposition in Speech-Transformer [29]. In addition, common model compression methods include knowledge distillation [30,31] and pruning operations [32], but knowledge distillation and pruning mostly require retraining of the model, and the training process is more complicated. To the best of our knowledge, most studies have focused on the feed-forward and convolutional layers in the network, and few studies have been conducted on the multi-head attention module, especially the multi-head self-attention module of the currently more advanced Conformer end-to-end speech recognition model. Therefore, in order to reduce the number of parameters and storage space in the Conformer model, we use low-rank decomposition to compress the multi-head self-attention module of the Conformer model for the study.

Speech recognition has been a relatively popular research object, but most research objects are mainly focused in resource-rich languages, such as Chinese datasets [33,34], English datasets [35] and other datasets [36,37], and very little work has been conducted to study speech recognition of languages in Central and West Asia. The Kazakh language is currently used in two regions: the first is Kazakhstan and Mongolia, in which the Cyrillic

alphabet is used; the second is the Xinjiang Uyghur Autonomous Region of China, in which the alphabet of Arabic origin is used [13]. The Kazakh language in the Cyrillic alphabet is used as a modeling unit in this paper. The Kazakh language of Cyrillic alphabet has 42 letters, some of which are rarely used; the letters В, Ё, Ц, Ч, Ъ, Ь, Э, Щ, $\phi$, һ, etc. only appear in loanwords. The research on Kazakh speech recognition is mainly focused on the following. Mamyrbayev et al. developed an end-to-end Kazakh speech recognition model based on RNN-T [38]. Orken et al. developed an end-to-end Kazakh speech recognition model based on Transformer [39]. Mamyrbayev et al. used different models for end-to-end agglutinative language (Kazakh) speech recognition for research training [40]. The Mamyrbayev et al. study built a hybrid end-to-end Kazakh speech recognition model based on BLSTM using CTC [41]. The above studies either only build a single speech recognition model for Kazakh, or the datasets they use are relatively difficult to acquire or are not open-source.

In addition, we found that the open-source Cyrillic Kazakh dataset suffers from uneven data distribution, i.e., high-frequency words contribute most of the training, while low-frequency words are rarely trained. As a result, the end-to-end speech recognition model always performs better recognition on high-frequency vocabulary, while the recognition accuracy decreases on low-frequency vocabulary, leading to the overall low recognition accuracy of the model. Recently, several approaches have been proposed in the field of speech recognition to solve the problem of uneven data distribution. The first, more common one is to improve the recognition of tail words by fusing language models in the end-to-end model. Toshniwal et al.'s study enables the model to recognize low-frequency words well when decoding by using a fusion language model [42–44]. Secondly, in overcoming the problem of uneven data distribution in multilingual speech recognition tasks, Winata et al. [45] attempted to improve the recognition rate of multilingual speech recognition by pre-training language models and using class priors to adjust the output of the softmax function. To alleviate the long-tail problem of single language in speech recognition, Deng et al. used a two-step training approach, i.e., representation learning and classification learning, in an end-to-end speech recognition model as a way to improve the recognition of low-frequency words by trying to add multiple loss functions (for example, by adding a softmax loss function with temperature in Transformer decoder) and pre-training the language model [46]. Previous work studies have not explored the long-tail problem in single small language speech recognition.

## 3. Methods

### 3.1. Low Rank Approximation Decomposition

We use a low-rank approximation algorithm based on singular value decomposition and apply it to the Conformer model to construct the LMHSA module. This section introduces the structure of our proposed low-rank Conformer model, which consists of a low-rank encoder and a low-rank decoder. Section 3.1.1 focuses on the low-rank encoder; Section 3.1.2 focuses on the low-rank decoder.

#### 3.1.1. Low-Rank Encoder

Unlike the Conformer encoder, our encoder uses a low-rank decomposition multi-head self-attention module to replace the multi-head self-attention module in the original model. Each encoding layer in the encoder consists of a stack of four modules: two feedforward modules, a low-rank decomposition multi-head self-attention module, and a convolution module. The detailed structure is shown in Figure 1.

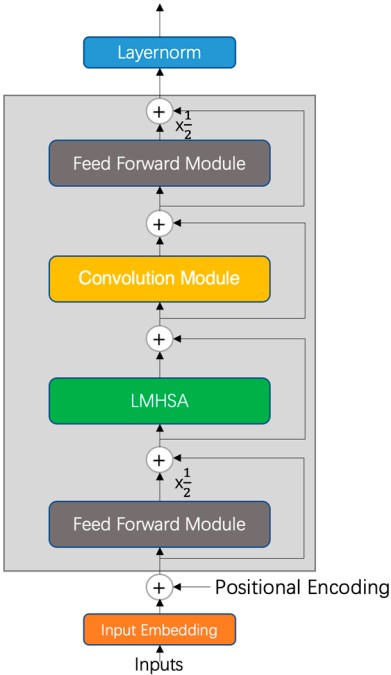

**Figure 1.** Low-rank encoder structure diagram.

When the *i*-th encoder layer input $X_i$, the output $Y_i$ can be expressed as follows:

$$\widetilde{x}_i = x_i + \frac{1}{2}FFN(x_i) \tag{1}$$

$$x'_i = \widetilde{x}_i + LMHSA\left(\widetilde{x}_i\right) \tag{2}$$

$$x''_i = x'_i + Conv\left(x'_i\right) \tag{3}$$

$$y_i = LayerNorm\left(x''_i + \frac{1}{2}FFN\left(x''_i\right)\right) \tag{4}$$

where *FFN*(·) denotes the feed-forward module, *LMHSA*(·) denotes the low-rank multi-head self-attention module, *Conv*(·) denotes the convolution module, and *LayerNorm*(·) denotes the layer normalization. Next, we focus on the low-rank multi-head self-attention module.

The multi-head self-attention mechanism maps *Q*, *K*, *V*, by linear transformations and then stitches the outputs together. The original multi-head self-attention module of Conformer's encoder is denoted as follows.

$$MultiHead(Q, K, V) = Concact(head_1, head_2, \ldots, head_n) \tag{5}$$

$$head_i = Attention(Q_i, K_i, V_i) \tag{6}$$

$$Q_i = QW_i^Q, K_i = KW_i^K, V_i = VW_i^V \tag{7}$$

To reduce the number of parameters in the model, we attempt to construct a low-rank multi-head self-attention module using a low-rank decomposition. We try to decompose the weight matrix of the multi-head self-attention module into the product of two matrices using low-rank approximation decomposition. The weight matrix *W* can be decomposed by a low-rank approximation as $\hat{U}D\hat{V}^T$, where $W \in \mathrm{R}^{m \times n}$, $\hat{U} \in \mathrm{R}^{m \times m}$, $D \in R^{m \times n}$, $\hat{V} \in R^{n \times n}$. Then, we proceed to decompose the matrix D into two identical

matrices $E$, i.e., $D = E^2$. If the parameters of the matrix are $m \times m$, the decomposed parameters are $m \times r + m \times r = 2mr$. If the rank r is less than m, the number of parameters after compression is less than the number of parameters of the original matrix. We use low-rank compression in the training and inference phases. The final decomposition result is shown in the following Equation (8):

$$W = UV^T \tag{8}$$

where $U = \hat{U}E$, and $V = E\hat{V}$. After applying the low-rank decomposition multi-head self-attended module, it can be expressed as:

$$LMultiHead(Q, K, V) = Concact\left(head'_1, head'_2, \dots, head'_n\right) \tag{9}$$

$$head'_i = Attention\left(Q'_i, K'_i, V^i_i\right) \tag{10}$$

$$Q'_i = QW_i^{Q'}, K'_i = KW_i^{K'}, V^i_i = VW_i^{V'} \tag{11}$$

where $W_i^{Q'}$, $W_i^{K'}$, and $W_i^{V'}$ are the decomposed matrix of $W_i^Q$, $W_i^K$ and $W_i^V$, respectively.

### 3.1.2. Low-Rank Decoder

The decoder consists of three main parts, two multi-head self-attention modules and a feed-forward network module, as shown in Figure 2.

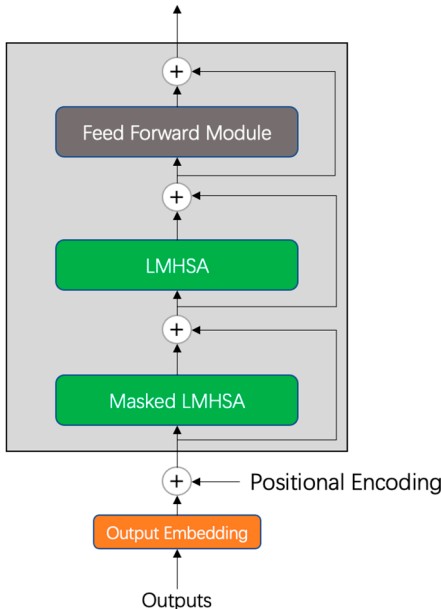

**Figure 2.** Low-rank decoder structure diagram.

Since the multi-head self-attention module of the decoder is relatively similar to that of the encoder, the low-rank decomposition of the decoder and the low-rank decomposition of the encoder have similar effects, which will not be elaborated upon in detail here.

### 3.2. Balanced Softmax

In the current end-to-end speech recognition, the softmax classification output is calculated as shown in the following equation:

$$p_{y_t} = \frac{\exp\left(f_{y_t}\right)}{\sum\limits_{y'_t \in [N]} \exp\left(f_{y'_t}\right)} \tag{12}$$

where $p_{y_t}$ is the output of softmax.

Applying Bayes' theorem reveals that the regular softmax is affected by the migration of the label distribution, and makes an estimate with a bias that causes the classifier computed by softmax regression to prefer that the sample belongs to the common class [47]. It is also for this reason that high-frequency words may have an impact on the tail words, making the model training a biased learning process.

Inspired by ref. [46], this article improves the output of the softmax function in the Conformer model and solves the problem of uneven data distribution by adding a penalty factor into the softmax classifier in the Attention model structure. The penalty factor is similar to the temperature in knowledge distillation [48]. The penalty factor is used to reduce the penalty for low-frequency words and the effect of high-frequency words on them, so that the classification of words is as smooth as possible, as a way to slow down the biased learning process of the model. The penalty factor is used in the training and inference phases, and is calculated as shown below.

$$p_{y_t} = \frac{\exp\left(\sigma f_{y_t}\right)}{\sum\limits_{y'_t \in [N]} \exp\left(\sigma f_{y'_t}\right)} \tag{13}$$

where $\sigma$ is a penalty factor, a hyperparameter greater than 1, and $p_{y_t}$ is the output of the softmax function after using the penalty factor.

### 3.3. Connectionist Temporal Classification

In sequence-to-sequence end-to-end speech recognition, the problem to be considered is the indefinite length between the input and output. In speech recognition, when the length of the output is smaller than the length of the input, the CTC [16] practice is to add blank labels that can be reused to align the output with the labels. As a result, there are many possibilities for the output. In order to efficiently handle the transformation relationship between the output and the label, CTC finally needs to remove the blank labels and optimize the objective as much as possible:

$$p(y \mid x) = \sum_{\pi \in \phi(y)} p(\pi \mid x) \tag{14}$$

where $\phi(y)$ denotes the set that can be converted into a sequence of corresponding labels by adding blank labels, inputting equal lengths, and finally, by removing blank labels. The final CTC loss calculation formula is then obtained:

$$L_{\text{CTC}} = -\ln(p(y \mid x)) \tag{15}$$

In the calculation of the path, CTC uses the forward–backward algorithm idea from HMM and the dynamic planning algorithm for calculation; the formula is as follows:

$$p(y \mid x) = \sum_u \frac{\alpha_t^u \beta_t^u}{q_t^{\pi t}} \tag{16}$$

where $\alpha_t^u$ and $\beta_t^u$ denote the sum of the forward and backward probabilities of label $u$ at moment $t$, respectively.

### 3.4. Efficient Conformer ASR

By adding a penalty factor to the softmax classifier in the Conformer model decoder, the loss of the Attention model is calculated as follows:

$$Att_{loss} = Att_{loss}(p, y) \tag{17}$$

where $p$ is the acoustic feature after the penalty factor is added, and $y$ is the corresponding label.

Since CTC can monotonically align the output and label sequences using blank labels and the forward–backward algorithm, the performance of the model can be further improved by adding CTC as a secondary task in the multitask learning framework, sharing the same encoder network with the Attention model, and using a joint multitask loss function. In this article, the performance of the optimized model is optimized using the joint training of CTC and Attention model structures, where the loss of the Attention model is computed after adding a penalty factor.

The loss function for multi-task learning is shown in Equation (18).

$$L_{\text{loss}} = \lambda CTC_{\text{loss}} + (1 - \lambda) Att_{\text{loss}} \tag{18}$$

where $\lambda$ is a hyperparameter, and $CTC_{\text{loss}}$ and $Att_{\text{loss}}$ represent the CTC loss and the attention loss after adding the penalty factor, respectively.

We construct an efficient Conformer model, including a low-rank encoder and a low-rank decoder, using a hybrid CTC/Attention multi-task training approach for joint training whose structure is shown in Figure 3. CTC consists of a linear layer and a logarithmic softmax layer. The CTC loss function is applied to the softmax output in training.

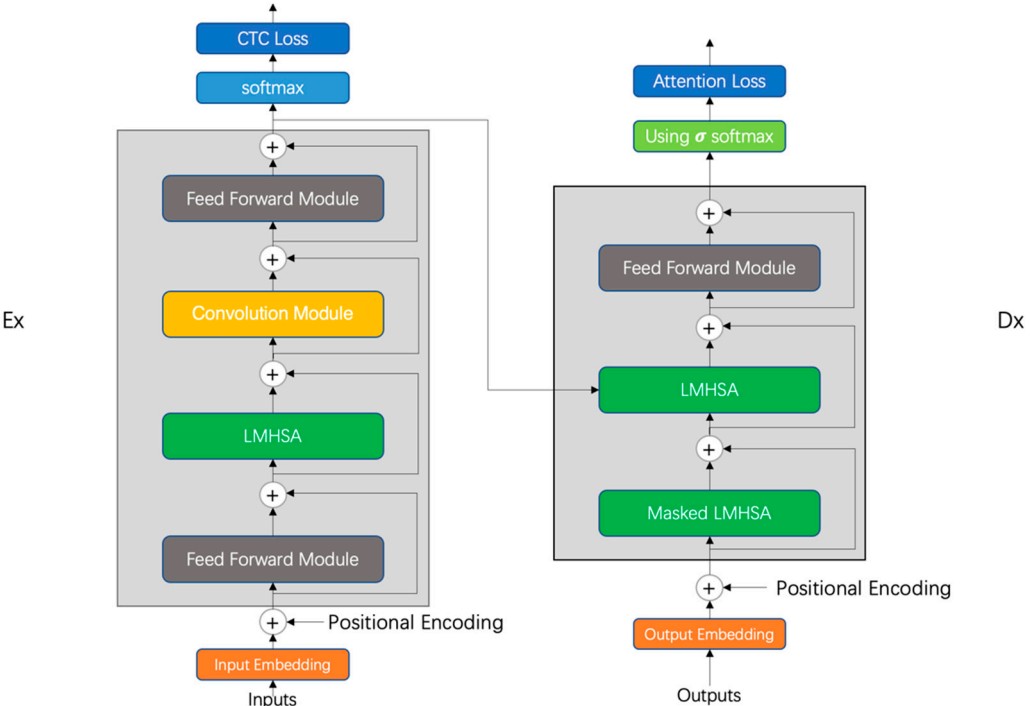

**Figure 3.** Illustration of our end-to-end Conformer framework graph using low-rank approximate decomposition and balanced softmax.

## 4. Experimental Environment

### 4.1. Dataset

The dataset used for the experiments is the Kazakh language dataset KSC from the open source [13]. The KSC dataset contains approximately 332 h of transcribed audio from different regions, ages, genders, recording devices, and various environmental conditions (e.g., home, office, and coffee shop, etc.). The content includes e-books, laws, and websites. All speech data are sampled at 16 kHz and quantized at 16 bit.

We counted the open-source Kazakh language dataset, and the distribution of its high-frequency words, low-frequency words, and the ratio of their frequencies are shown in Table 1.

**Table 1.** Like [46]-style long-tail data distribution in the Kazakh language dataset.

| | KSC |
|---|---|
| Training set | KSC-train |
| Utterance | 147,236 |
| Tail word number | 152,566 |
| Head word number | 4625 |
| Tail word frequency | 558,919 |
| Head word frequency | 1,044,443 |
| Frequency ratio of tail and head | 65.2:34.8 |
| Development set | KSC-dev |
| Utterance | 3283 |
| Tail word frequency | 11,288 |
| Head word frequency | 21,730 |
| Frequency ratio of tail and head | 65.8:34.2 |
| Testing set | KSC-test |
| Utterance | 3334 |
| Tail word frequency | 13,505 |
| Head word frequency | 21,830 |
| Frequency ratio of tail and head | 61.7:38.2 |

As can be seen from Table 1, the number of head categories (high-frequency words) accounts for a small percentage compared to the number of tail categories (low-frequency words), but their frequency of occurrence is high, and the ratio of high-frequency words to low-frequency words is about 65%.

*4.2. Experimental Configuration*

In the experiments of this article, the input of the model is an 80-dimensional Fbank feature, in which the frame length is 25 ms and the frame shift is 10 ms. We use a BPE subword to train transcribed text, with several special symbols such as <unk>, <sos>, <eos> and a blank tag, etc. The subword size is 2000, and Pytorch is used as the experimental platform for experiment.

The number of attention heads is 4, the number of position feed-forward units is 2048, the dropout [49] of each encoder layer is set to 0.1, the output dimension is 256, and the number of convolutional kernels in the convolutional network is set to 15, using a 12-layer Conformer encoder. The Adam optimizer was used for training, the learning rate was set to 0.0004, and the warm-up step was 25,000. Transformer's decoder is six layers with four attention heads, and each layer contains 2048 units. During training, the weight parameter of the CTC model is set to 0.3, where the penalty factor is set to 1.1. The experiments in this paper use masking of partial information in the time and frequency domains during training to expand the dataset, where the masking parameter F is 10, and T is 50 [50]. The maximum number of iterations in the decoding process is 50, and the batch size is set to 16. In this article, the optimal value of the last ten checkpoints at the end of training is used as the decoding model. A single card, a GPU NVIDIA V100 32GB graphics card, is used in this paper. The experimental results were obtained without an additional language model.

The other comparative experimental configurations are as follows. In the traditional HMM speech recognition model and BLSTM speech recognition model, we refer to [13], cited for use. The DNN-HMM model uses the Kaldi framework with input MFCC features, where the frame length is 25ms and the frame shift is 10ms. The BLSTM-CTC model encoder uses a three-layer bidirectional LSTM network with 1024 nodes in each layer. The decoder uses LSTM and CTC, where the CTC weight is 0.5 and each layer of the LSTM network contains 1024 nodes. The maximum number of iterations is 20. No data enhancement technique is used during training. For the Transformer model, we use 12 layers of encoder and 6 layers of decoder, where the input is 80-dimensional Fbank features, the number of attention heads is 4, the hidden layer dimension is 256, the feed-forward location network

dimension is 2048, and the dropout is set to 0.1. The weight of CTC during training is 0.3. The maximum number of iterations is 50.

## 5. Results and Discussion

In order to prove the recognition performance of the method and model proposed in this article, we carried out several sets of experiments. Firstly, in the low-rank decomposition experiments, we compared the traditional model and the end-to-end model, and then analyzed the best rank value, comparing the content of the recognition accuracy, the number of parameters and the model size. Secondly, in the long-tail problem processing, we compare with the mainstream models and explore the softmax using the best penalty factor value in the Conformer model, testing the criteria for the accuracy of recognition. Finally, we explored the experiments related to the efficient Conformer model. The word error rate is calculated as shown in the following equation.

$$WER = \frac{I + D + R}{N} \tag{19}$$

where *I* is the insertion error, *D* is the deletion error, *R* is the replacement error, and *N* is the total number of tags within the real tag sequence.

### 5.1. Low-Rank Decomposition Experiment

We constructed the Conformer low-rank multi-head self-attention encoder and decoder using low-rank decomposition. Table 2 is the comparison between the compressed low-rank decomposition and other models.

**Table 2.** Comparison of low rank compression and each model.

| Model | Number of Parameters | WER |
|:---:|:---:|:---:|
| Traditional hybrid structure | | |
| DNN-HMM [13] | - | 13.7% |
| End-to-end architecture | | |
| BLSTM-CTC [13] | - | 28.8% |
| Transformer-CTC | 31.7 M | 12.85% |
| Baseline (Conformer-CTC) | 47.6 M | 10.36% |
| Baseline + low rank(r = 128) | 47.6 M | 10.33% |
| Baseline + low rank(r = 64) | 44.1 M | 10.85% |
| Baseline + low rank(r = 32) | 42.3 M | 11.00% |

As shown in Table 2, Conformer has a larger number of parameters compared to the other models, which is 47.6 M. When the lightweight Conformer model is constructed using a low-rank decomposition, the number of parameters of the model decreases. When the rank is 128, the number of parameters of the model does not decrease. When the rank is 64, the number of model parameters decreases by 3.5 M. When the rank is 32, the number of model parameters decreases by 5.3 M. It is also observed that the word error rate of the model increases as the number of parameters is compressed and reduced.

The following table shows the comparison between the Conformer model and the baseline model after decomposing the Conformer model with different rank sizes.

As can be seen in Table 3, the size of the model decreases as the rank value decreases. When the rank is 128, although the storage size of the model does not change, the word error rate of the model decreases and is reduced by 0.03% compared to the baseline model. As the compression rank value decreases, the recognition rate of the model decreases. This also implies that the size of the rank value cannot be set too low, otherwise it will have a greater impact on the model recognition results.

**Table 3.** We used Conformer as the baseline model and then compared the decomposition results of different rank sizes.

| Model | Size | ΔSize | ΔWER |
|---|---|---|---|
| Baseline (Conformer-CTC) | 182 MB | 0 | 0 |
| Baseline + low rank (r = 128) | 182 MB | 0 | 0.03 |
| Baseline + low rank (r = 64) | 168.5 MB | −13.5 MB | −0.49% |
| Baseline + low rank (r = 32) | 161.8 MB | −20.2 MB | −0.64% |

Figure 4 shows the train loss for the Conformer baseline model and the low-rank decomposition Conformer model using different size ranks. It can be seen that the loss of the model after using the low-rank decomposition has increased to different degrees compared to the baseline model.

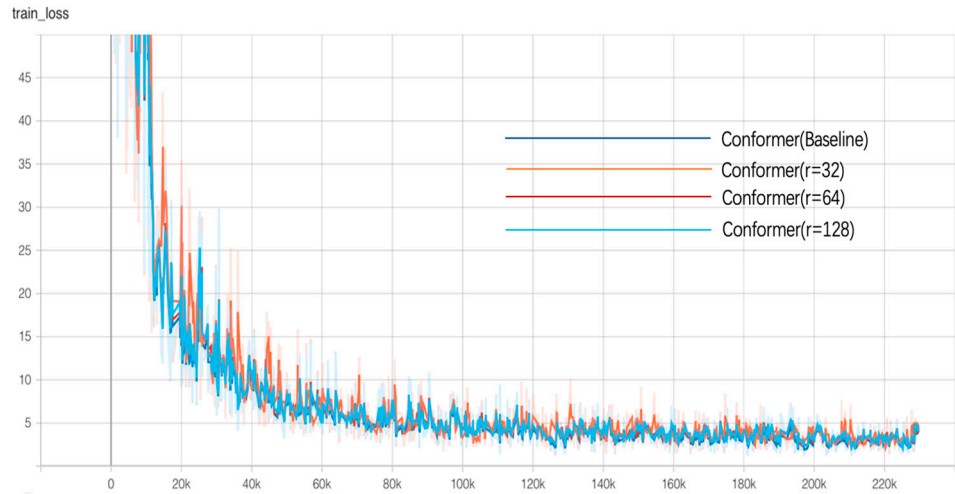

**Figure 4.** Training loss of different rank values of Conformer model and baseline model.

*5.2. Experiment with Balanced Softmax*

In order to prove the effectiveness of the improved model after using the balanced softmax function, we compare with the current mainstream models. Table 4 shows the recognition results of the Kazakh language dataset under different models.

**Table 4.** Comparison experiments between the Conformer model using balanced softmax and other models.

| Model | WER |
|---|---|
| DNN-HMM [13] | 13.7% |
| BLSTM-CTC [13] | 28.8% |
| Transformer-CTC | 12.85% |
| Conformer no CTC | 15.38% |
| Baseline (Conformer-CTC) | 10.36% |
| Baseline + balanced softmax | 10.21% |

From the results in Table 4, it can be seen that the performance results of BLSTM-CTC are not as good as the recognition results of the traditional DNN-HMM model, with a 15.1% difference in results. The Transformer model, on the other hand, using a multi-head self-attention mechanism, results in a good pair of results compared to the first two, with a relative decrease in the word error rate of 0.85% and 15.95%. The Conformer model enhances the Transformer's learning of local features by using a convolutional module, and its recognition results achieve a good result compared to the Transformer; the result is reduced by 2.49%. The model with the balanced softmax function has a decrease compared to the baseline model, with a 0.15% decrease in the word error rate.

The reason for this is that adding a penalty factor can reduce the penalty for low-frequency words, enhance the model's learning of low-frequency words, and thus improve the overall recognition performance.

Table 5 gives the comparative recognition results of some low-frequency words after using the penalty factor. For example, in the Kazakh test dataset, the word Кесектерді (translated as: pieces) appears with a frequency of one, which is a low-frequency word. When the training model does not use the penalty factor, the word is incorrectly recognized as Кезектерді, while when the model uses the improved method, the word is correctly recognized.

**Table 5.** Before and after comparison experiments of some low-frequency words using balanced softmax. Translation * using online translation.

| Word in KSC | Translation * | Number of Appearances | Baseline | Use balanced softmax |
|---|---|---|---|---|
| егде | older | one | екіде | егде |
| пазл | puzzle | one | пазыл | пазл |
| инеге | to the needle | one | енеге | инеге |
| Күпті | a lot | one | Күтті | Күпті |
| жұлын | spinal cord | one | жолын | жұлын |
| Кесектерді | pieces | one | Кезектерді | Кесектерді |
| қызу | hot | one | қызыл | қызу |

The loss of the training process of the model is shown in Figure 5. It can be seen that using CTC as an auxiliary function in the multi-task learning approach can accelerate the convergence of the model and speed up the training, reduce the loss value in the model training, and improve the recognition effect of the model.

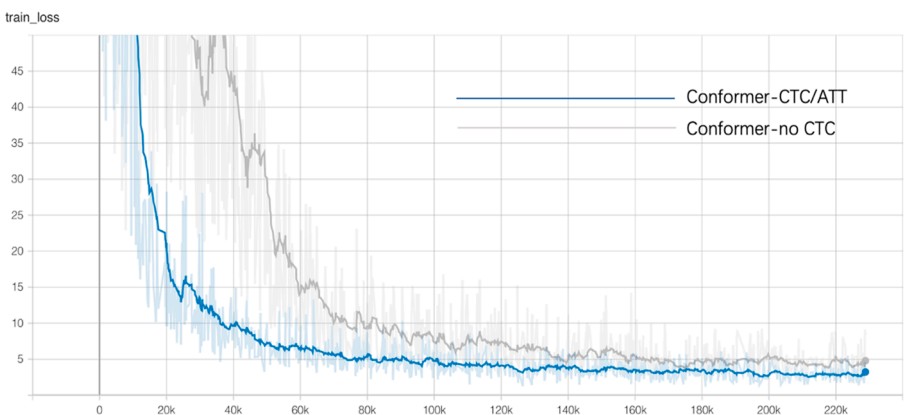

**Figure 5.** The train loss with and without CTC as an auxiliary task.

To verify the effect of setting different sizes of penalty factors on the model recognition results, we use Conformer as the baseline model and conduct comparative experiments with models using different sizes of penalty factors.

From Table 6, it can be seen that adding the penalty factor can effectively improve the recognition effect; compared with the baseline model, the penalty factor causes the word error rate to decrease by 0.15%. However, the penalty factor cannot be set too large, otherwise the recognition accuracy will be reduced.

**Table 6.** Results for different sizes of penalty factor in the baseline Conformer model, where 1 represents no penalty factor used.

| Different Size Penalty Factor | WER |
|:---:|:---:|
| 1 | 10.36% |
| 1.1 | 10.21% |
| 1.2 | 10.25% |

In order to explore the effect on the model when the penalty factor is set at different positions, we use a penalty factor value of 1.1 as the baseline model in this article and conduct a comparison experiment with the model, with the penalty factor set at different positions. The experimental results are shown in Table 7.

**Table 7.** Experiments using a penalty factor with a value of 1.1 in the training or inference phase, and their combination.

| Position | WER |
|:---:|:---:|
| Training + Inference phases | 10.21% |
| Training | 10.34% |
| Inference phases | 10.34% |

As can be seen from Table 7, the penalty factors set at different positions do have some effect on the model. None of the recognition results set in the training or inference phases separately are better than baseline with the penalty factor set in both the training and inference phases.

### 5.3. Efficient Conformer Model Using Low Rank Decomposition and Balanced Softmax

Through the above experiments, we combine low-rank decomposition and a balanced softmax function to construct an efficient low-rank decomposition Conformer model, compressing the number of parameters and size of the model as much as possible while the word error rate and training time are basically unchanged. The experimental results are shown in Table 8.

**Table 8.** Results of the ablation study of the proposed Conformer architecture.

| Model | Parameters | Size | WER |
|:---:|:---:|:---:|:---:|
| Baseline | 47.6 M | 182 MB | 10.36% |
| Baseline + low rank compression | 44.1 M | 168.5 MB | 10.85% |
| Baseline + balanced softmax | 47.6 M | 182 MB | 10.21% |
| Baseline + low rank compression + balanced softmax | 44.1 M | 168.5 MB | 10.54% |

As can be seen from Table 8, the efficient Conformer model consists of the low-rank encoder and the low-rank decoder. Without increasing the training difficulty, the effective Conformer improves only 0.18% WER in comparison to the baseline model; however, the parameters of the model are reduced by 7.4% and the storage space is reduced by 13.5 MB.

Figure 6 shows the comparison of the training loss of our proposed efficient Conformer model and the baseline Conformer model. It can be seen that the loss of our proposed efficient Conformer model is basically unchanged from the baseline Conformer model.

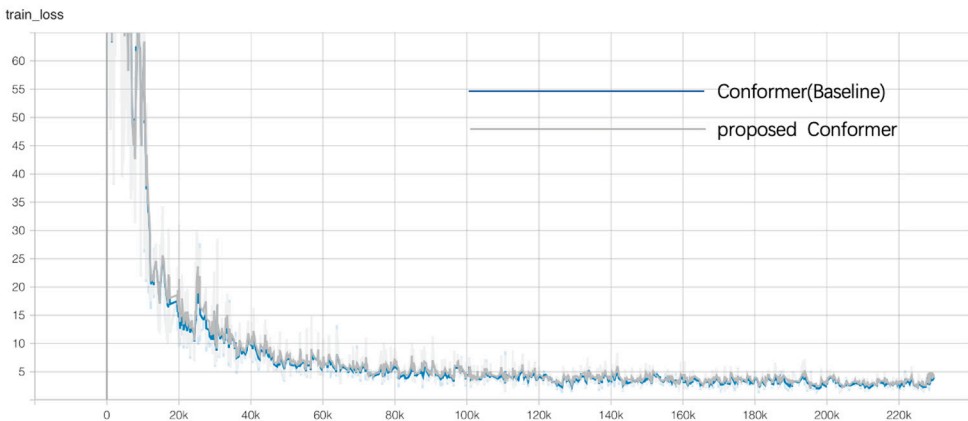

**Figure 6.** The train loss of the proposed efficient Conformer model and the baseline Conformer model.

## 6. Conclusions and Future Work

We proposed an efficient Conformer model based on low-rank decomposition. We construct a low-rank multi-head self-attention module of the encoder and decoder to reduce the number of parameters and storage space of the multi-head self-attention module, and do not need to retrain the model when it has been compressed; we use the balanced softmax function to replace the original softmax function to alleviate the biased effect of high-frequency words on low-frequency words in the Kazakh language dataset, so as to improve the recognition accuracy. The final number of parameters of the model is compressed by 7.4%, and the storage space of the model is compressed by 13.5 MB; we validated the effectiveness of the model structure through ablation comparison experiments, and the training time and word error rate are basically unchanged. In future work, we will try to further compress the model using low-rank decomposition to compress the feed-forward and convolutional layers, while using methods such as model quantization to further improve the training speed and recognition process.

**Author Contributions:** Writing—original draft, T.G.; writing—review and editing, N.Y. and W.S. All authors have read and agreed to the published version of the manuscript.

**Funding:** This work was supported by the National Natural Science Foundation of China—Research on Key Technologies of Speech Recognition of Chinese and Western Asian Languages under Resource Constraints (Grant No. 62066043); the National Language Commission key Project: ZDI135-133.

**Institutional Review Board Statement:** Not applicable.

**Informed Consent Statement:** Not applicable.

**Data Availability Statement:** Not applicable.

**Conflicts of Interest:** The authors declare no conflict of interest.

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
