# Peer review of "Efficient Conformer for Agglutinative Language ASR Model Using Low-Rank Approximation and Balanced Softmax"

_applsci, doi:10.3390/app13074642_

Round 1

Reviewer 1 Report

The topic of the work could be of interrest for the readers in the area of speech processing, and there is some kind of novelty in the combination ot the low rank MHSA, balanced softmax and the hybrid CTC/attention for ASR. The paper is well structured and clearly written, the work is technically sound and the experiments and conclusiones are significant.

However, there are some problem which should be fixed:

- The last part of section 3.1.1 on low-rank encoder should be rewritten, mainly because of the notation. From the expression (7), the authors use a general wright matrix W, but they should particularize for every head i and the queries Q, keys K and values V. Then they refer to expression (13) for the final decomposition, but this expression is not about that. Then the matrix E is not defined. Then the they do not explain how they implement the lowering of the rank, without defining the 'r' parameter used in the experimental section. They do not mention if it is done for every instance, only in training, only in testing or both training and testing. Then expressions (9) and (10) are identical to (5) and (6), respectively. It should be fixed with the appropriate notation. The low-ranked matrixes cannot have named as the original ones. The notation of expression (11) should be algo changed, adding the index i to all matrixes to the U and V matrixes. All of these problems should be fixed.

- In section 3.2 it should be clear the novelty of the penalty factor of expression (13) in relationship with references [46] and [48]

- In section 3.3, references are missing. And more improtant, more details are necessary about the integration of the CTC in the Conformer, and about the computation of the expression (16)

- In section 4.2 is not clear how the input embedding is computed from the spectrogram

- English is in general good, but it should be reviewed. For instance, in expression 

Author Response

I am very pleased to have your review and thank you very much for pointing out my mistakes in detail. I have corrected my article as you requested. Once again, thank you very much for your review.

For the first point, I reintroduced the compression process in detail and introduced r and its in training and reasoning stages, and finally corrected the formula as you meant.
For the second point, I reintroduced the method. Because Balanced Softmaxs is inspired by reference [46], the penalty factor is similar to the function of reference [48].
For the third point, I added the reference and introduced the use of CTC in section 3.4, because I think this is more consistent with the overall presentation of the model, and if you have other comments you can point them out to me and I will improve them immediately.
To the fourth point, I added the introduction of subwords.
I hope to get your reply.

Reviewer 2 Report

This paper examines end-to-end speech recognition using the Conformer framework, for some Central and West Asian agglutinative languages.  The work constructed a low-rank multi-head self-attention encoder and decoder using low-rank approximation decomposition, and used CTC as an auxiliary task to speed up the model training.  It conducted experiments on the open source Kazakh language dataset.

The paper is reasonable.  I found many small errors, as listed below, but the paper is acceptable.

Specific changes needed:

CTC is used several times before being defined in section 2.

..of Central and West Asian agglutinative language.  ->

..of Central and West Asian agglutinative languages. 

..and consume more resources. ->

..and it consumes more resources.

..For these reason, we ..

..For these reasons, we ..

..Automatic speech recognition (ASR), also known as speech recognition, .. ->

..Automatic speech recognition (ASR)..

..[2]proposed ..

..[2] proposed ..

(Put a space before and after each [..], unless immediately followed by a punctuation mark such as comma or period)

..English dataset [3]compared .. ->

..English datasets [3] compared ..

..Conformer end-to-end model have ..

..the Conformer end-to-end models have ..

..model[7]. li et al.

..model [7].  Li et al.

..[9] use two ..

..[9] uses two ..

..decomposition(SVD) ..

..decomposition (SVD) ..

..of DNN.. Low-rank ..

..of DNN.  Low-rank ..

..[10,11], for example, ..

..[10,11]; for example, ..

..The long-tail problem has .. - explain what this problem is, when the term is first introduced, not several sentences later

..To our best know, ..

..To our best knowledge, ..

..an lightweight but ..

..a lightweight but ..

(This error occurs more than once)

..results on the KSC dataset .. - give details of this

..Balanced Softmax function .. - give details of this

..rate(WER) ..

..rate (WER) ..

..transformer uses a ..

..Transformer uses a ..

..Kriman et al.proposed ..

..Kriman et al. proposed ..

..Mehrotra et al.proposed ..

..Mehrotra et al. proposed ..

..Mamyrbayev et al. study built .. 

..The Mamyrbayev et al. study built ..

..Cyrillic Kazakh is a small language .. - small in what sense?

..multiple loss functions(Such as ..

..multiple loss functions (such as ..

..section 3.1.1 focuses on ..

..Section 3.1.1 focuses on ..

..Includes: two feedforward modules, a low-rank decomposition multi-head self attention module and a convolution module.  - not necessary to include this, as is clear in the figure (same comment for figure 2)

..where FFN(-) denotes .. - do not indent this, right after an equation (same after eqs. 8, 11, 12, 14, 16)

..needs to remove the blank label ..

..needs to remove the blank labels ..

..u at moment t, .. - be consistent in use of italics for math symbols (same comment for p and y in the ensuing text)

..Other comparative experimental configurations.  - this is not a sentence

..no data enhancement ..

..No data enhancement ..

..Transformer model we use ..

..For the Transformer model, we use ..

.. is 0.3. the maximum ..

.. is 0.3.  The maximum ..

.. the number of parameter and ..

.. the number of parameters and ..

..Where the word error rate is calculated as shown in the following equation.  - this is not a sentence

..parameter of Conformer have a larger number of parameter .. - rephrase; poor

..When the rank is 128, .. - the text in this section is too repetitive, with the same same sentence structure in successive sentences; perhaps use a table instead

..15.95%. the Conformer ..

..15.95%.  The Conformer ..

..Transformer the result is .. 

..Transformer; the result is ..

..that adding penalty factor ..

..that adding a penalty factor ..

What is the use of * in table 5?

The sentence at the end of page 13: “As can be seen from Table 8, ..” Is too long, and with too many commas

..the number of parameter and ..

..the number of parameters and ..

..In Librispeech: an asr corpus ..

..Librispeech: an ASR corpus ..

(Also: no need to repeat “IEEE, 2015”.. twice in the same reference; far too many references repeat the year, “IEEE”, ICASSP, etc)

Too many references have the useless word “In”

Be consistent in placement of the year in references.

Author Response

I am very pleased to have your review and thank you very much for pointing out my mistakes in detail. I have corrected my article as you requested. Once again, thank you very much for your review.
